# UniL: Point Cloud Novelty Detection through Multimodal Pre-training

Yuhan Wang
College of Software Engineering, Southeast University
Nanjing, Jiangsu, China
Key Laboratory of New Generation Artificial Intelligence
Technology and Its Interdisciplinary Applications
(Southeast University), Ministry of Education
Nanjing, Jiangsu, China
eugene_wong@seu.edu.cn

Mofei Song*
School of Computer Science and Engineering, Southeast
University
Nanjing, Jiangsu, China
Key Laboratory of New Generation Artificial Intelligence
Technology and Its Interdisciplinary Applications
(Southeast University), Ministry of Education
Nanjing, Jiangsu, China
songmf@seu.edu.cn

## Abstract

3D novelty detection plays a crucial role in various real-world applications, especially in safety-critical fields such as autonomous driving and intelligent surveillance systems. However, existing 3D novelty detection methods are constrained by the scarcity of 3D data, which may impede the model's ability to learn adequate representations, thereby impacting detection accuracy. To address this challenge, we propose a Unified Learning Framework (UniL) for facilitating novelty detection. During the pretraining phase, UniL assists the point cloud encoder in learning information from other modalities, aligning visual, textual, and 3D features within the same feature space. Additionally, we introduce a novel Multimodal Supervised Contrastive Loss (MSC Loss) to improve the model's ability to cluster samples from the same category in feature space by leveraging label information during pretraining. Furthermore, we propose a straightforward yet powerful scoring method, Depth Map Error (DME), which assesses the discrepancy between projected depth maps before and after point cloud reconstruction during novelty detection. Extensive experiments conducted on 3DOS have demonstrated the effectiveness of our approach, significantly enhancing the performance of the unsupervised VAE method in 3D novelty detection. Codes are avaliable at https://github.com/EugeneWon9/UniL.

## CCS Concepts

• **Computing methodologies** → **3D imaging**; **Anomaly detection**.

## Keywords

Point Cloud, Novelty Detection, Multimodal Pre-training, Supervised Contrastive Learning

*Corresponding author

**ACM Reference Format:**
Yuhan Wang and Mofei Song. 2024. UniL: Point Cloud Novelty Detection through Multimodal Pre-training. In *Proceedings of the 32nd ACM International Conference on Multimedia (MM '24), October 28-November 1, 2024, Melbourne, VIC, Australia.* ACM, New York, NY, USA, 10 pages. https://doi.org/10.1145/3664647.3680988

## 1 Introduction

Novelty detection, also known as "novel class detection"[33, 34], primarily focuses on identifying semantic shifts. It aims to detect any test samples that do not fall into any training category, as the term "novel" generally refers to the unknown, new, and something interesting[63]. Distributional shifts compared to the training data can significantly impact model performance, posing potential threats or risks. For instance, in safety-critical applications such as autonomous driving systems[25, 65], the model's ability to detect and reject unknown samples becomes crucial, as it must return control to the driver. In the realm of 2D analysis, the field of novelty detection has reached a relatively advanced stage. However, within the 3D domain, despite significant advancements in visual understanding[13, 16, 22, 26, 30, 31, 57], novelty detection remains nascent and has received little attention from researchers. Antonio et al.[3] present the first benchmark for 3D Open Set learning (3DOS), which includes various tasks of increasing difficulties regarding semantic shifts and encompasses both in-domain and cross-domain scenarios. Building on this foundation, the authors of 3DOS conducted an extensive survey of methods for out-of-distribution (OOD) detection and Open Set recognition across 2D and 3D domains, evaluating them on the proposed benchmark.

However, we observed a significant limitation in those methods: the scarcity of available 3D data. Compared to 2D data, acquiring and annotating 3D data is typically more expensive and time-consuming, resulting in a limited availability of annotated datasets. This limitation can lead to inadequately trained models that may struggle to represent all possible scenarios and variations. Moreover, the increased complexity of feature extraction from 3D data may cause suboptimal representation, thus further impacting the model's performance in novelty detection.

To address this challenge, we opted to integrate information from other modalities of point cloud data, transferring knowledge from pre-trained multimodal models. Among the methods evaluated on

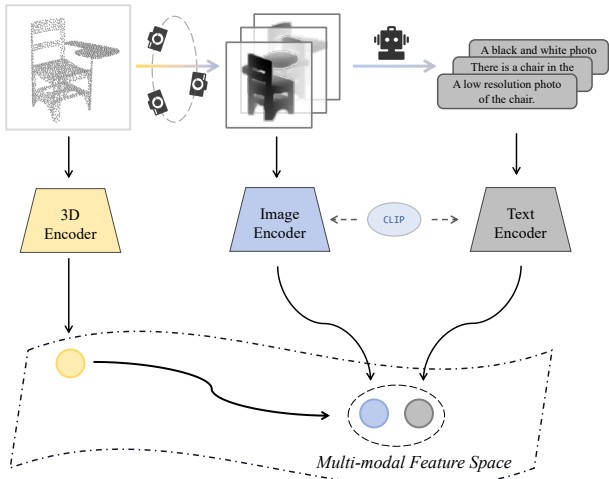

**Figure 1: The visualization of UniL. To address the issue of insufficient data in 3D novelty detection tasks, UniL adopts a pretraining approach to allow the 3D encoder to learn more information. By aligning point cloud, depth map, and text features simultaneously, the 3D encoder can achieve better performance in subsequent novelty detection task.**

3DOS[3], only VAE[35] is unsupervised, making it our chosen baseline. Specifically, our approach consists of three phases: pretraining the 3D encoder, training the VAE, and performing novelty detection. These phases are illustrated in Fig. 1, which outlines the workflow of our approach. Firstly, to tackle the challenge of limited 3D data, we incorporate information from both text and image domains, constructing a Unified Learning Framework that facilitates the 3D encoder in assimilating knowledge from these modalities, called UniL. To further enhance the alignment accuracy in the feature space, we designed the Multimodal Supervised Contrastive Loss (MSC Loss), which fully utilizes the label information of point cloud data. This enables the 3D encoder to focus more on the similarity among samples from the same category, forming tighter clusters in the feature space. Secondly, during the VAE training phase, we employ the pre-trained encoder and utilize its encoded features to fit the mean and variance. After reparameterization sampling, the features are input into the decoder to reconstruct the input point cloud. Finally, to uncover underlying novel patterns, we integrate information from the 2D domain and introduce the Depth Map Error (DME). The computation of DME involves projecting the reconstructed point cloud onto a depth map and calculating the error between it and the original depth map. By comparing the depth maps before and after reconstruction, we can quantify the quality of the point cloud reconstruction and extract potential novel patterns. This approach not only compensates for the shortcomings of limited 3D data but also fully utilizes the correlations between different modalities, thereby improves the robustness and accuracy of novelty detection. After thorough evaluation on the benchmark introduced in 3DOS[3], the effectiveness of our approach has been demonstrated. Specifically, our method achieved top-1 performance on SN1 (hard), with a notable improvement of 2.0% in AUROC and a reduction of 5.5% in FPR95. On SR1 (easy), we observed a reduction in FPR95 of 0.9%, while on SR1 (hard), we achieved a notable improvement in AUROC of 1.9%.

Our main contributions are summarized as follows:

- To address the issue of insufficient 3D data, we constructed a Unified Learning Framework (UniL) to assist the 3D encoder in learning knowledge from other modalities.
- We proposed Multimodal Supervised Contrastive Loss (MSC Loss), which utilizes label information to further enhance the alignment accuracy of features across different modalities.
- In order to incorporate information from different modalities for novelty detection, we introduced Depth Map Error (DME), a simple yet efficient approach that detects potential novelties by evaluating the projection error of depth maps both before and after point cloud reconstruction.

## 2 Related Work

### 2.1 3D Point Cloud Learning

There are mainly two streams for learning from point cloud. One is projecting point clouds into voxels[36, 49] or images[24, 66] and then using 2D/3D convolutions for feature extraction, given the irregular and unordered nature of point cloud structures. The other one is directly processing point cloud data, with PointNet[41] being the first neural network to adopt this method. It can effectively learn and extract features from unordered point sets, which significantly influences point-based 3D networks. DGCNN[59] proposed a dynamic graph structure for performing convolution operations on unordered point sets. PointMLP[32] is a simple feed-forward residual MLP network that hierarchically aggregates the local features extracted by MLPs while eliminating the need for intricate local geometric extractors.

Moreover, recent work has adopted self-supervised pretraining methods for 3D understanding. Point-BERT[67] encodes point cloud data into text sequences and applies self-attention mechanism to learn the semantic relationships and representations between points. Point-MAE[39] directly processes the point cloud by masking out 3D patches and then predicting them back using L2 loss. PointGPT[7] applies the principles of GPT[44] to the generation of point cloud data, acquiring robust 3D representations through pretraining on autoregressive generation tasks. While these methods have shown effectiveness, their full potential remains unrealized due to the limited availability of 3D data.

### 2.2 Multimodal Pre-Training

Multimodal pretraining aims to learn universal representations by simultaneously processing multiple types of data, such as images, text, speech, etc. This paradigm helps models understand the semantic correlations between different modalities, thereby achieving better performance in various downstream tasks.

CLIP[43] trains the model by contrastive learning between images and text, allowing natural language to understand visual concepts. As the first successful multimodal learning model, CLIP holds significant importance in advancing research on multimodal learning and understanding the semantic relationships between images and text. Inspired by CLIP-based adaption methods[12, 28, 68], PointCLIP V2[69] and CLIP2Point[18] convert point clouds into 2D forms through projection and rendering, and then apply the powerful generalization capability of CLIP to zero-shot classification.

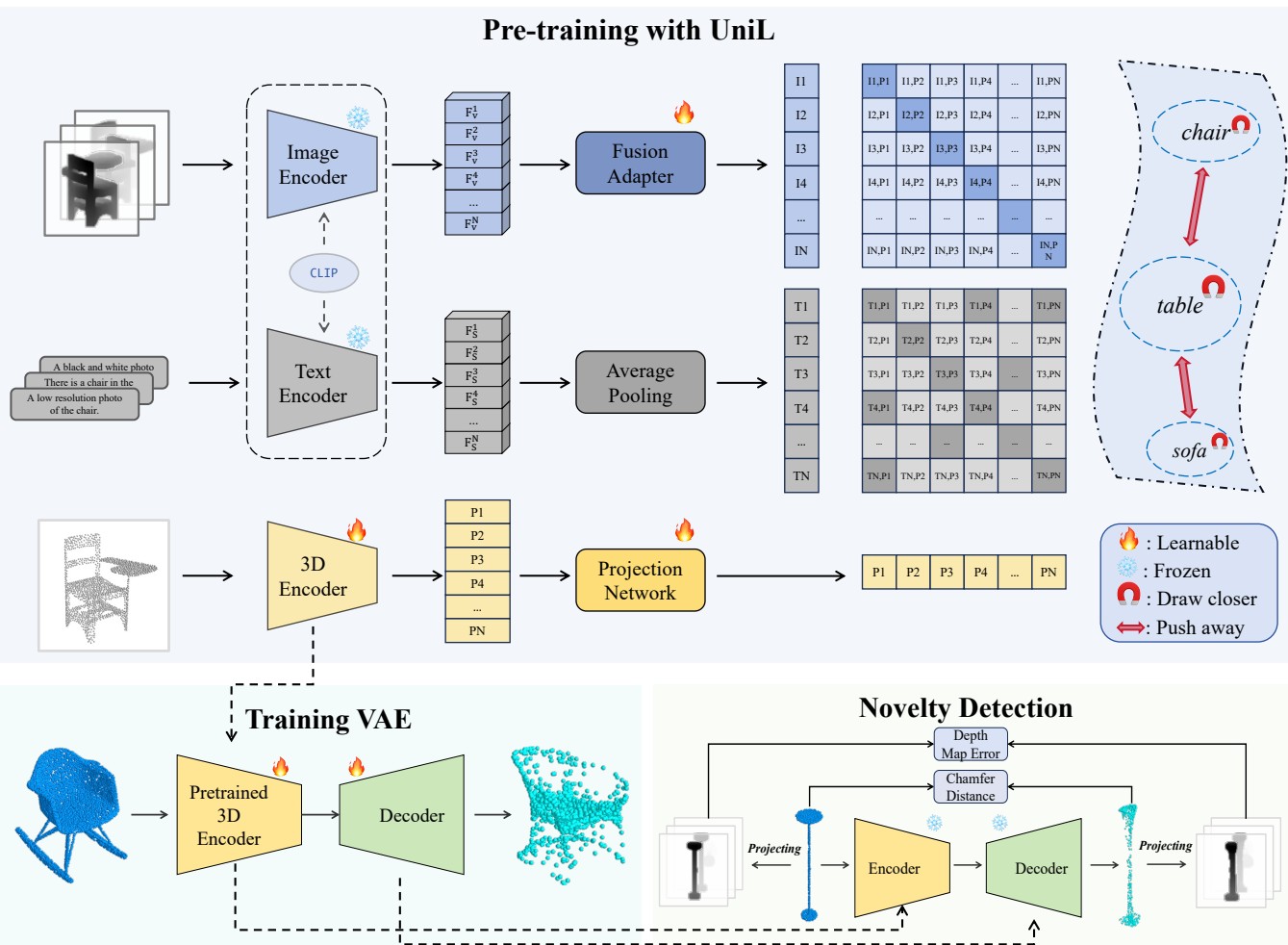

**Figure 2: The framework of our method. During the pretraining phase (Top) of the 3D encoder using the Unified Learning Framework (UniL), depth maps of point clouds and textual information are input into the frozen CLIP. The Multimodal Supervised Contrastive Loss (MSC Loss) utilizes the label information of point clouds to enhance the compactness of 3D representations from the same category in the feature space. Subsequently, in the VAE training phase (Bottom Left), the pretrained 3D encoder is loaded and fine-tuned. Finally, Depth Map Error (DME) is employed during novelty detection (Bottom Right) to compare depth maps of point clouds before and after reconstruction, enabling the identification of hidden novel patterns.**

ULIP[61] aims to directly adapt the paradigm of CLIP to learn a unified representation space for point clouds, language, and images.

Drawing inspiration from ULIP, we aimed to incorporate the information of point cloud in other modalities, enhancing the representation capabilities of the 3D encoder. Unlike ULIP, which neglects the semantic similarity of point cloud data in the language modality, we introduce the Multimodal Supervised Contrastive Loss (MSC Loss) to refine feature alignment accuracy.

## 2.3 Novelty Detection

Currently, novelty detection methods primarily focus on density-based, classification-based, distance-based, and reconstruction-based approaches. Among these, density-based methods[1, 9, 23, 40, 48, 70] identify novel samples by measuring the density around the data points, since novel samples typically reside in sparse regions. Classification methods[45, 47, 52, 58] attempt to assign test samples to predefined categories, while samples that cannot be classified can be considered as novel samples. As novelties are assumed to be distant from the training data, distance-based methods[37, 51, 53] assess novelty by computing the distance between a test sample and the training samples. In reconstruction-based methods[4, 8, 21, 38, 54, 62], the model's ability to reconstruct known patterns is leveraged to detect data points that significantly differ from the training data. However, these methods are primarily designed for traditional 2D data, and therefore perform poorly when transferred to the 3D domain.

[35] marks the pioneering attempt to address anomaly detection in 3D point clouds of general objects, introduces an framework built upon a variational autoencoder.Following the work of [35], [2] proposed a novel unsupervised approach for detecting novelty in 3D point cloud, utilizing a general feature extractor for point clouds and a one-class classifier.

Considering that these methods rely solely on 3D data for detection, we propose the Depth Map Error (DME) to assess the novelty of point clouds in the 2D domain.

## 3 Method

In this section, we will provide a detailed exposition of our method. Firstly, we pretrain the 3D encoder within the UniL framework, enhancing feature alignment accuracy through MSC Loss. Following this, we load the pretrained 3D encoder and train the VAE to reconstruct input point cloud samples. Lastly, during the novelty detection phase, we introduce additional 2D visual information and employ DME to measure the disparity between the projected depth maps of point clouds before and after reconstruction. The overall framework is illustrated in Fig. 2.

### 3.1 Pre-training with UniL

Limited data may impose constraints on the model's ability to learn meaningful and accurate feature representations, thereby impacting its performance and generalization capability. To address this issue, we adopt a multimodal pretraining approach. UniL leverages rich information from data in various modalities and transfers it to 3D encoder by aligning multimodal features within a unified feature space, enhances the encoder's learning capacity and improves feature representation.

*3.1.1* ***Data Preparation***. To obtain a unified semantic space integrating point clouds, vision, and language, we followed ULIP and created a dataset consisting of point clouds, images, and textual descriptions. For each CAD model indexed as $i$, we constructed a triplet sample $T_i$ consisting of a point cloud $P_i$, a depth map collection $\{D_v^i\}_{v=1}^N$, and a text description set $S_i$. After uniformly sampling $N$ points from the original point cloud of CAD model, we applied standard point cloud data augmentation techniques, including random point drop, scaling, shifting, and rotation.

Next, we choose the Realistic Projection proposed in [69] instead of the rendering to obtain depth maps of point clouds, as depth maps can more accurately reflect the geometric shapes and spatial structures of objects. Specifically, 3D voxel grid $G \in \mathbb{R}^{H \times W \times D}$ is assigned different depth values to represent the depth information of the point cloud $P_i = (x, y, z)$ by utilizing:

$$G(\lceil sHx \rceil, \lceil sWy \rceil, \lceil Dz \rceil) = z, \tag{1}$$

where $H$, $W$, $D$ denote the spatial resolutions of G and $s \in (0, 1]$ is a scale factor. Following this, local minimum pooling operations are applied to densify the grid, and the depth values are reassigned to occupy previously sparse voxels. Next, a non-parametric Gaussian kernel is employed for the purposes of smoothing and filtering, compressing the depth dimension to produce the final projected depth map. We project each point cloud from N different viewpoints to generate a collection of depth maps denoted as $\{D_v^i\}_{v=1}^N$ for point cloud $p_i$.

Finally, following ULIP's work, we utilize the category name of each point cloud during pretraining, using 63 prompts such as "an image of [category]" and an additional prompt "a point cloud model of [category]" to accommodate the 3D modality. At the pretraining stage, a set of textual descriptions $S_i$ is generated by applying 64 templates to the category name of each point cloud.

*3.1.2* ***Feature Extraction***. In pretraining, acquiring high-quality representations from other modalities is crucial, as it necessitates aligning the features of the 3D modality with these representations to gain knowledge from other modalities. We choose CLIP[43] as the teacher for our 3D encoder and freeze the parameters of its visual and textual encoders. By leveraging CLIP's reliable representations, we bridge the gap between the limited availability of 3D data and the rich multimodal knowledge encapsulated in CLIP.

For the visual modality, we input each depth map of $\{D_v^i\}$ into the visual encoder $E_i(\cdot)$ of CLIP, and obtain the depth feature collection for sample i as follows:

$$\{F_v^i\}_{v=1}^N = E_i(\{D_v^i\}_{v=1}^N). \tag{2}$$

Drawing from the insights of [18], we furthur employ the proposed Adapter to dynamically fuse the depth feature set $\{F_v^i\}_{v=1}^N$ across multiple views. Subsequently, the image-domain feature $f_i^I$ for sample $i$ can be obtained via the following formulation:

$$f_i^I = f_2(ReLU(f_1(\sum_{v=1}^N \alpha_v \cdot F_v^i))), \tag{3}$$

where $f_1$ and $f_2$ represent two-layer MLP networks, and $\alpha_v$ denotes the dynamic fusion coefficients for the $v$-th view.

For the linguistic modality, we employ CLIP's text encoder $E_i(\cdot)$ to process the descriptions $S_i$, generating a set of text representations. Following this, we employ average pooling on the resulting set, yielding the text-domain feature representation $f_i^T$ for each sample $i$, expressed as:

$$f_i^T = AVG(E_t(S_i)), \tag{4}$$

where the $AVG(\cdot)$ denotes the average pooling operation.

Within the VAE framework, the decoder receives features that have been re-parameterized. Recognizing that aligning features across different training paradigms may lead to unstable training and potential information loss, we choose to directly utilize the global features obtained from the encoder. To convert the encoded 3D feature into a multimodal embedding space, we incorporate a projection network after the 3D encoder $E_p(\cdot)$. This allows us to formulate the final 3D features $f_i^P$ as follows:

$$f_i^P = Proj(E_p(P_i)), \tag{5}$$

where $P_i$ is the augmented point cloud, and $Proj(\cdot)$ is a single-layer MLP.

*3.1.3* ***Multimodal Representation Alignment***. The objective of pretraining is training the 3D point cloud encoder $E_p(\cdot)$ to align the 3D features of sample $i$ with its image and text features. Thus, ULIP investigates the feasibility of transferring 2D contrastive learning to 3D domain by adopting a contrastive loss similar to CLIP to achieve alignment between 3D and image features:

$$\mathcal{L}_{P2I} = -\frac{1}{2}(\sum_i log \frac{exp(f_i^P f_i^I/\tau)}{\sum_j exp(f_i^P f_j^I/\tau)} + \sum_i log \frac{exp(f_i^P f_i^I/\tau)}{\sum_j exp(f_j^P f_i^I/\tau)}), \tag{6}$$

where $i$ and $j$ are indices of samples, and $\tau$ is a learnable temperature parameter. Similarly, the alignment between 3D and text features is formulated as:

$$\mathcal{L}_{P2T} = -\frac{1}{2}(\sum_i log \frac{exp(f_i^P f_i^T/\tau)}{\sum_j exp(f_i^P f_j^T/\tau)} + \sum_i log \frac{exp(f_i^P f_i^T/\tau)}{\sum_j exp(f_j^P f_i^T/\tau)}). \quad (7)$$

Although the contrastive loss used in ULIP is relatively effective in aligning feature representations from different modalities, simply maximizing the logits on the main diagonal of the similarity matrix between 3D and text is not sufficiently accurate. This is because point clouds from the same category share identical prompts, and these prompts, after being encoded by the frozen CLIP, yield identical textual features. Hence, inspired by the supervised contrastive learning[19], we propose a novel Multimodal Supervised Contrastive Loss (MSC Loss), which modifies the original self-supervised 3D-to-text loss to:

$$\mathcal{L}_{P2T}^{sup} = -\frac{1}{2}\Big( \sum_i \sum_j 1_{\tilde{y}_i = \tilde{y}_j} \cdot \log \frac{exp(f_i^P f_j^T/\tau)}{\sum_k exp(f_i^P f_k^T/\tau)} + \quad (8)$$
$$\sum_j \sum_i 1_{\tilde{y}_j = \tilde{y}_i} \cdot \log \frac{exp(f_i^P f_j^T/\tau)}{\sum_k exp(f_k^P f_i^T/\tau)} \Big)$$

where $i$, $j$, $k$ are indices of the samples, $\tilde{y}$ represents the label information of the samples, and $\tau$ is a temperature parameter. This loss function indicates that 3D features belonging to the same class are brought closer together in the embedding space, while simultaneously pushing apart 3D features from different classes. Therefore, the ultimate training objective is to train the 3D encoder $E_p(\cdot)$, minimizing the MSC Loss:

$$\mathcal{L}_{MSC} = \mathcal{L}_{P2I} + \mathcal{L}_{P2T}^{sup}. \quad (9)$$

Through Multimodal Supervised Contrastive learning, the 3D encoder can acquire more discriminative feature representations, leading to samples from same category being closer while scattering those of different categories further apart.

## 3.2 Training VAE

During the training of the VAE, known categories of point clouds are provided as inputs. By minimizing the disparity between the original data and the reconstructed data, the VAE can discern patterns and structures in normal data. Consequently, when confronted with unseen samples, their reconstruction error tends to be notably higher compared to that of normal samples.

### 3.2.1 Point Cloud Reconstruction.
Following the setup of [3], we first randomly sample $N$ points from the original point cloud, and then apply 3D data augmentation techniques such as random rotation, jitter, translation, and scaling. Prior to training, we load the pretrained point cloud encoder aforementioned and fine-tune it during the subsequent training process. Each sample $P_i$ is then passed through the point cloud encoder $E_p$, yielding the global feature $f_i^{global}$ of the point cloud, from which we estimate the mean $\mu_i$ and variance $\sigma_i$ as follows:

$$f_i^{global} = E_p(p_i), \quad (10)$$
$$\mu_i = f_1(f_i^{global}), \quad (11)$$
$$\sigma_i = f_2(f_i^{global}), \quad (12)$$

where $f_1$ and $f_2$ are single-layer FC layers. Next, we sample from the distribution $\mathcal{N}(\mu_i, \sigma_i^2)$ to obtain the reparameterized feature $Z_i$:

$$Z_i = \mu_i + \varepsilon \cdot \sigma_i, \quad (13)$$

where $\varepsilon$ is sampled from $\mathcal{N}(0, 1)$. Finally, $Z_i$ is fed into the decoder $D_p$ inspired by [64], to obtain the reconstructed point cloud $\hat{p}_i$:

$$\hat{p}_i = D_p(Z_i). \quad (14)$$

### 3.2.2 Training Objective.
Existing literature has introduced two permutation-invariant metrics for comparing unordered point sets: Chamfer Distance and Earth Mover's Distance[11, 46]. Following the setup of [35], we utilize CD to compute the reconstruction error $L_{rec}$ for point cloud sample $P_i$. This choice is motivated by the faster convergence and lower computational cost of CD compared to EMD. $L_{rec}$ is calculated as follows:

$$L_{rec} = \sum_i \Big( \sum_{x \in P_i} \min_{\hat{x} \in \widehat{P}_i} ||x - \hat{x}||_2 + \sum_{\hat{x} \in \widehat{P}_i} \min_{x \in P_i} ||\hat{x} - x||_2 \Big) \quad (15)$$

Following the traditional VAE[20] approach, the KL divergence is utilized as the fitting loss to minimize the discrepancy between the Gaussian distribution $\mathcal{N}(0, 1)$ and $\mathcal{N}(\mu, \sigma^2)$ derived from the original point cloud $P_i$. The KL divergence is defined as:

$$D_{KL\ ori} = D_{KL}(\mathcal{N}(\mu, \sigma^2)||\mathcal{N}(0, 1)). \quad (16)$$

Additionally, [35] employ a second KL divergence to measure the difference between the Gaussian distribution $\mathcal{N}(0, 1)$ and $\mathcal{N}(\hat{\mu}, \hat{\sigma}^2)$ obtained by inputting the reconstructed point cloud $\hat{P}_i$ into the network:

$$D_{KL\ rec} = D_{KL}(\mathcal{N}(\hat{\mu}, \hat{\sigma}^2)||\mathcal{N}(0, 1)). \quad (17)$$

Therefore, the overall training objective is defined as:

$$\mathcal{L} = L_{rec} + D_{KL\ ori} + D_{KL\ rec}. \quad (18)$$

## 3.3 Novelty Detection

To assess whether a sample is novelty, [35] adapt the reconstruction error calculated using Chamfer Distance as its anomaly score. However, VAE is a versatile generative model that samples from a latent space to generate data. Since the latent space is continuous, even if the input data is unseen, the model might find similar points in the latent space and decode them to generate reconstructions similar to the input data, resulting in a relatively small reconstruction error.

To tackle this challenge, we introduce the Depth Map Error (DME) as a score to incorporate information from other modalities of the point cloud. DME quantifies the disparity in depth maps between the original and reconstructed point clouds. This approach aims to mitigate the aforementioned challenge, providing a more effective measure to capture novelties. Therefore, the score used for novelty detection can be formalized as:

$$Score = w \cdot CD(P_i, \hat{P}_i) + (1 - w) \cdot DME(\{D_v^i\}_{v=1}^N, \{\hat{D}_v^i\}_{v=1}^N), \quad (19)$$

where $DME(\cdot)$ represents the function computing the mean squared error between pixels of two depth maps, and $w$ is the scoring coefficient.

**Table 1: Results on the Synthetic Benchmark track. Each column title indicates the chosen known class set, the other two sets serve as unknown. "DGC" refers to the backbone network DGCNN[59], while "PN2" denotes the backbone network PointNet++[42].**

| | Synthetic Benchmark | | | | | | | | | | | | | | | | | |
| Method | SN1(hard) | | SN2(med) | | SN3(easy) | | Avg | | Method | SN1(hard) | | SN2(med) | | SN3(easy) | | Avg | |
| | AUROC↑ | FPR95↓ | AUROC↑ | FPR95↓ | AUROC↑ | FPR95↓ | AUROC↑ | FPR95↓ | | AUROC↑ | FPR95↓ | AUROC↑ | FPR95↓ | AUROC↑ | FPR95↓ | AUROC↑ | FPR95↓ |
|---|---|---|---|---|---|---|---|---|---|---|---|---|---|---|---|---|---|
| DGC+MSP[14] | 74.0 | 83.9 | 88.6 | 62.4 | 92.9 | 43.2 | 85.2 | 63.2 | PN2+MSP[14] | 74.3 | 82.8 | 80.0 | 78.1 | 89.7 | 52.2 | 81.3 | 71.0 |
| DGC+MLS[56] | 75.1 | 77.7 | 91.1 | 42.6 | 92.4 | 35.2 | 86.2 | 51.8 | PN2+MLS[56] | 72.0 | 80.8 | 83.9 | 64.1 | 89.8 | 40.5 | 81.9 | 61.8 |
| DGC+ODIN[27] | 75.4 | 76.5 | 91.1 | 42.9 | 92.5 | 34.4 | 86.3 | 51.3 | PN2+ODIN[27] | 74.2 | 79.4 | 79.4 | 71.7 | 87.8 | 41.8 | 80.5 | 64.3 |
| DGC+Energy[29] | 75.2 | 77.0 | 91.2 | 41.6 | 92.3 | 36.4 | 86.2 | 51.7 | PN2+Energy[29] | 72.1 | 81.2 | 84.0 | 64.7 | 89.8 | 39.4 | 82.0 | 61.8 |
| DGC+GradNorm[17] | 66.2 | 88.1 | 80.9 | 64.0 | 71.6 | 77.7 | 72.9 | 76.6 | PN2+GradNorm[17] | 72.1 | 81.8 | 57.7 | 88.9 | 57.8 | 79.0 | 62.6 | 83.3 |
| DGC+ReAct[50] | 76.4 | 74.6 | 92.5 | 37.9 | 96.4 | 19.3 | 88.4 | 43.9 | PN2+ReAct[50] | 73.7 | 79.4 | 89.6 | 52.1 | 95.0 | 27.2 | 86.1 | 52.9 |
| DGC+OE+mixup[15] | 73.7 | 78.9 | 90.4 | 44.7 | 91.4 | 46.0 | 85.2 | 56.5 | PN2+OE+mixup[15] | 72.7 | 78.9 | 80.3 | 68.8 | 87.3 | 62.2 | 80.1 | 69.9 |
| DGC+ARPL+CS[6] | 72.9 | 84.2 | 90.7 | 47.1 | 89.5 | 89.5 | 84.4 | 73.6 | PN2+ARPL+CS[6] | 74.8 | 80.3 | 80.7 | 72.4 | 85.4 | 50.8 | 80.3 | 67.8 |
| DGC+Cosine Proto | 84.3 | 59.1 | 88.8 | 39.7 | 86.4 | 48.0 | 86.5 | 48.9 | PN2+Cosine Proto | 80.3 | 68.3 | 88.7 | 60.8 | 91.9 | 38.0 | 86.9 | 55.7 |
| DGC+CE($L^2$) | 80.4 | 75.5 | 90.1 | 40.9 | 96.7 | 14.4 | 89.1 | **43.6** | PN2+CE($L^2$) | 83.4 | 66.8 | 89.5 | **37.7** | 92.9 | 28.1 | 88.6 | 44.2 |
| DGC+SupCon[19] | 80.3 | 75.7 | 84.6 | 73.6 | 87.9 | 44.3 | 84.3 | 64.5 | PN2+SupCon[19] | 80.9 | 75.5 | 83.5 | 68.2 | 85.1 | 45.1 | 83.2 | 62.9 |
| DGC+SubArcFace[10] | 81.2 | 73.4 | **91.9** | 44.0 | 94.9 | 26.5 | **89.3** | 48.0 | PN2+SubArcFace[10] | 79.0 | 81.2 | 82.9 | 60.3 | 89.1 | 32.8 | 83.7 | 58.1 |
| DGC+NF | 82.0 | 74.8 | 86.1 | 53.8 | **97.4** | **11.5** | 88.5 | 46.7 | PN2+NF | 81.5 | 72.5 | 71.1 | 78.0 | 91.0 | 49.6 | 81.2 | 66.7 |
| VAE[35] | 67.2 | 76.9 | 69.5 | 83.4 | 94.3 | 32.4 | 77.0 | 64.2 | VAE(Ours) | **86.3** | **53.6** | 80.9 | 77.4 | 96.5 | 20.6 | 87.9 | 50.5 |

**Table 2: Results on the Synthetic to Real Benchmark track. Each column title indicates the chosen known class set, the other two sets serve as unknown. "DGC" refers to the backbone network DGCNN[59], while "PN2" denotes the backbone network PointNet++[42].**

| | Synth to Real Benchmark | | | | | | | | | | | | | |
| method | SR1(easy) | | SR2(hard) | | Avg | | method | SR1(easy) | | SR2(hard) | | Avg | |
| | AUROC↑ | FPR95↓ | AUROC↑ | FPR95↓ | AUROC↑ | FPR95↓ | | AUROC↑ | FPR95↓ | AUROC↑ | FPR95↓ | AUROC↑ | FPR95↓ |
|---|---|---|---|---|---|---|---|---|---|---|---|---|---|
| DGC+MSP[14] | 72.2 | 91.0 | 61.2 | 90.3 | 66.7 | 90.6 | PN2+MSP[14] | 81.0 | 79.6 | 70.3 | 86.7 | 75.6 | 83.2 |
| DGC+MLS[56] | 69.0 | 92.2 | 62.4 | 88.9 | 65.7 | 90.5 | PN2+MLS[56] | 82.1 | 76.6 | 67.6 | 86.8 | 74.8 | 81.7 |
| DGC+ODIN[27] | 69.0 | 92.2 | 62.4 | 89.0 | 65.7 | 90.6 | PN2+ODIN[27] | 81.7 | 77.3 | 70.2 | 84.4 | 76.0 | 80.8 |
| DGC+Energy[29] | 68.8 | 92.7 | 62.4 | 88.9 | 65.6 | 90.8 | PN2+Energy[29] | 81.9 | 77.5 | 67.7 | 87.3 | 74.8 | 82.4 |
| DGC+GradNorm[17] | 67.0 | 93.5 | 59.8 | 89.4 | 63.4 | 91.5 | PN2+GradNorm[17] | 77.6 | 80.1 | 68.4 | 86.3 | 73.0 | 83.2 |
| DGC+ReAct[50] | 68.4 | 92.1 | 62.8 | 88.8 | 65.6 | 90.5 | PN2+ReAct[50] | 81.7 | 75.6 | 67.6 | 87.2 | 74.6 | 81.4 |
| DGC+OE+mixup[15] | 71.1 | 89.6 | 59.5 | 92.0 | 65.3 | 90.8 | PN2+OE+mixup[15] | 71.2 | 89.7 | 60.3 | 93.5 | 65.7 | 91.6 |
| DGC+ARPL+CS[6] | 71.5 | 90.2 | 62.8 | 89.5 | 67.1 | 89.8 | PN2+ARPL+CS[6] | **82.8** | 74.9 | 68.0 | 89.3 | 75.4 | 82.1 |
| DGC+Cosine Proto | 58.6 | 90.6 | 57.3 | 91.3 | 57.9 | 91.0 | PN2+Cosine Proto | 79.9 | 74.5 | **76.5** | **77.8** | 78.2 | **76.1** |
| DGC+CE($L^2$) | 67.5 | 87.4 | 64.6 | 91.0 | 66.1 | 89.2 | PN2+CE($L^2$) | 79.7 | 84.5 | 75.7 | 80.2 | 77.7 | 82.3 |
| DGC+SubArcFace[10] | 74.5 | 86.7 | 68.7 | 86.6 | 71.6 | 86.7 | PN2+SubArcFace[10] | 78.7 | 84.3 | 75.1 | 83.4 | 76.9 | 83.8 |
| DGC+NF | 72.5 | 81.6 | 70.2 | 83.0 | 71.3 | 82.3 | PN2+NF | 78.0 | 84.4 | 74.7 | 84.2 | 76.4 | 84.3 |
| VAE[35] | 68.6 | 77.0 | 57.9 | 92.3 | 63.3 | 84.6 | VAE(Ours) | 72.2 | **73.6** | 62.6 | 92.1 | 67.4 | 82.9 |

**Table 3: Results on the Real to Real Benchmark track. Each column title indicates the chosen unknown class set, the other two sets serve as known. "DGC" refers to the backbone network DGCNN[59], while "PN2" denotes the backbone network PointNet++[42].**

| | Real to Real Benchmark | | | | | | | | | | | | | | | | | |
| Method | SR3(easy) | | SR2(med) | | SR1(hard) | | Avg | | Method | SR3(easy) | | SR2(med) | | SR1(hard) | | Avg | |
| | AUROC↑ | FPR95↓ | AUROC↑ | FPR95↓ | AUROC↑ | FPR95↓ | AUROC↑ | FPR95↓ | | AUROC↑ | FPR95↓ | AUROC↑ | FPR95↓ | AUROC↑ | FPR95↓ | AUROC↑ | FPR95↓ |
|---|---|---|---|---|---|---|---|---|---|---|---|---|---|---|---|---|---|
| DGC+MSP[14] | 83.0 | 69.4 | 72.0 | 88.7 | 57.5 | 90.3 | 70.8 | 82.8 | PN2+MSP[14] | 88.1 | 67.3 | 80.6 | 84.0 | 73.7 | 80.3 | 80.8 | 77.2 |
| DGC+MLS[56] | 84.9 | 58.2 | 79.0 | 81.0 | 54.0 | 92.8 | 72.6 | 77.3 | PN2+MLS[56] | 89.4 | 53.8 | **83.4** | 73.1 | 76.4 | **75.3** | 83.0 | 67.4 |
| DGC+ODIN[27] | 84.9 | 58.2 | 79.0 | 80.9 | 54.0 | 92.8 | 72.6 | 77.3 | PN2+ODIN[27] | 90.2 | 47.9 | 83.3 | 71.7 | 76.3 | 76.8 | 83.3 | 65.5 |
| DGC+Energy[29] | 84.8 | 59.7 | 79.1 | 81.4 | 53.8 | 93.2 | 72.6 | 78.1 | PN2+Energy[29] | 89.5 | 50.6 | 81.6 | 75.8 | 76.6 | 75.5 | 82.6 | 67.3 |
| DGC+GradNorm[17] | 77.5 | 73.3 | 73.3 | 87.4 | 51.0 | 92.9 | 67.2 | 84.5 | PN2+GradNorm[17] | 88.5 | 50.7 | 77.4 | 75.3 | 75.2 | 76.8 | 80.4 | 67.6 |
| DGC+ReAct[50] | 87.6 | 54.0 | 79.0 | 78.6 | 58.9 | 93.1 | 75.1 | 75.3 | PN2+ReAct[50] | 90.3 | 48.9 | 82.4 | 75.8 | 75.4 | 77.6 | 82.7 | 67.4 |
| DGC+OE+mixup[15] | 76.8 | 77.8 | 74.9 | 87.2 | 57.6 | 89.9 | 69.8 | 85.0 | PN2+OE+mixup[15] | 72.6 | 83.5 | 72.0 | 88.5 | 62.5 | 87.8 | 69.0 | 86.6 |
| DGC+Cosine Proto | 90.0 | 43.7 | 78.5 | 75.3 | 65.5 | 85.7 | 78.0 | 68.2 | PN2+Cosine Proto | **91.0** | **41.0** | 82.1 | 78.2 | 77.6 | 75.6 | **83.6** | 64.9 |
| DGC+CE($L^2$) | 83.1 | 59.3 | 74.5 | 77.2 | 67.1 | 86.8 | 74.9 | 74.4 | PN2+CE($L^2$) | 85.1 | 64.4 | 78.9 | 83.9 | 73.2 | 79.1 | 79.1 | 75.8 |
| DGC+SubArcFace[10] | 86.7 | 58.5 | 78.4 | 76.1 | 65.0 | 84.0 | 76.7 | 72.9 | PN2+SubArcFace[10] | 87.1 | 61.3 | 78.9 | 76.9 | 73.7 | 81.4 | 79.9 | 73.2 |
| DGC+NF | 76.9 | 77.3 | 71.7 | 82.7 | 61.8 | 86.2 | 70.2 | 82.1 | PN2+NF | 88.0 | 47.7 | 80.6 | **68.2** | 75.6 | 81.4 | 81.4 | 65.8 |
| VAE[35] | 56.4 | 88.8 | 55.8 | 90.6 | 52.3 | 99.1 | 54.8 | 92.8 | VAE(Ours) | 90.1 | 63.8 | 72.2 | 78.1 | **79.5** | 97.4 | 80.6 | 79.8 |

## 4 Experiment

To illustrate the advantages of utilizing the pretrained 3D encoder through UniL, we conducted experiments on 3DOS. First, we outline the experimental settings, including datasets and implementation details. Following this, we present the quantitative results of 3D novelty detection on three benchmarks. Finally, we conduct an analysis and demonstration of the proposed components, validating their effectiveness.

## 4.1 Datasets

We use the following dataset employed for building the benchmark proposed in 3DOS[3]:

**ShapeNetCore** [5] contains of 51,127 CAD models from 55 common object categories. In 3DOS, ShapeNetCore v2 is used with the official training (70%), validation (10%), and testing (20%) splits. All point clouds are uniformly sampled from mesh surfaces, normalized to fit within a unit cube centered at the origin and consistently

aligned. In the benchmark setting, similar semantic categories, like cellphone and telephone, are merged to obtain 54 classes.

**ModelNet40** [60] comprises 12,311 CAD models from 40 categories. 3DOS adopts the dataset partitioning of [42], which consists of 9,843 training samples and 2,468 testing samples. Each point cloud is uniformly sampled from the surfaces of synthetic CAD models and scaled to fit within a unit cube centered at the origin.

**ScanObjectNN** [55] is a dataset of scanned 3D objects from the real world, unlike ShapeNetCore and ModelNet40, consists of 2,902 samples from 15 categories. These samples can be divided into two categories based on the presence of background: *OBJ_ONLY* and *OBJ_BG*. In 3DOS, the original *OBJ_BG* split is considered, wherein 3D scans are impacted by acquisition artifacts. These samples are already in the form of point clouds, with each containing 2048 points, representing both foreground and background objects along with other interacting elements.

## 4.2 Implementation Details

*4.2.1* **Pre-training with UniL**. We uniformly sample 2048 points from each sample and generate projected depth maps and textual descriptions. CLIP is utilized to acquire multimodal embedding, while both the image and text encoder in our experiment remain frozen, akin to ULIP. During pretraining, only the parameters in the 3D encoder and projection network are trainable. We train Unil for 150 epochs with a batch size of 64 and a learning rate of 1e-4. The optimizer used is AdamW, with a weight decay of 0.1.

*4.2.2* **Training VAE**. During VAE training, we used publicly available code provided by [35], a practice also adopted by 3DOS[3]. Specifically, we randomly sampled 2048 points from each point cloud and applied data augmentation techniques including scaling, translation transformations, and random rotation around the up-axis. The Adam optimizer was employed with a weight decay of 1e-6, and the learning rate was set to 1e-3. Training was conducted with a batch size of 100 for 300 epochs.

In the *Synthetic Benchmark*, due to the relatively abundant data of ShapeNetCore, we fine-tuned the entire 3D encoder. However, in the *Synth to Real Benchmark*, where the source domain data distribution differs from that of the target domain, updating the encoder was found to lead to catastrophic forgetting. Hence, we opted to freeze the entire encoder. For the *Real to Real Benchmark*, due to the limited data of ScanObjectNN and to prevent overfitting, we froze the low-level semantic layers of the encoder and only updated the high-level semantic layers.

## 4.3 Novelty Detection

*4.3.1* **Benchmarks**. 3DOS assesses the capability to detect unknown samples in test data using the AUROC and FPR95 metrics. It includes three benchmarks: the *Synthetic Benchmark* is designed for scenarios involving only semantic shift; the more challenging *Synth to Real Benchmark* encompasses both semantic and domain shift, using train and test samples from synthetic data (ModelNet40) and real-world data (ScanObjectNN) respectively; the *Real to Real Benchmark* presents an intermediate scenario involving semantic shift between training and test data, accompanied by noisy samples from ScanObjectNN in both sets. Considering tasks with varying difficulty levels, the merged ShapeNetCore is divided into three

**Table 4: The ablation study of MSC Loss and DME on Synthetic Benchmark.**

| ULIP Loss | MSC Loss | DME | SN1(hard) | | SN2(med) | | SN3(easy) | | Avg | |
|---|---|---|---|---|---|---|---|---|---|---|
| | | | AUROC↑ | FPR95↓ | AUROC↑ | FPR95↓ | AUROC↑ | FPR95↓ | AUROC↑ | FPR95↓ |
| ✗ | ✗ | ✗ | 67.2 | 76.9 | 69.5 | 83.4 | 94.3 | 32.4 | 77.0 | 64.2 |
| ✓ | ✗ | ✗ | 70.8 | 77.6 | 73.6 | 81.5 | 95.1 | 23.1 | 79.8 | 60.7 |
| ✗ | ✓ | ✗ | 68.7 | 76.1 | 76.4 | 76.3 | 96.3 | **15.3** | 80.5 | 55.9 |
| ✓ | ✗ | ✓ | **86.6** | **52.2** | 78.5 | 84.8 | 95.8 | 22.4 | 87.0 | 53.1 |
| ✗ | ✓ | ✓ | 86.3 | 53.6 | **80.9** | 77.4 | **96.5** | 20.6 | **87.9** | **50.5** |

**Table 5: The ablation study of MSC Loss and DME on Synth to Real Benchmark.**

| ULIP Loss | MSC Loss | DME | SR1(easy) | | SR2(hard) | | Avg | |
|---|---|---|---|---|---|---|---|---|
| | | | AUROC↑ | FPR95↓ | AUROC↑ | FPR95↓ | AUROC↑ | FPR95↓ |
| ✗ | ✗ | ✗ | 68.6 | 77.0 | 57.9 | 92.3 | 63.3 | 84.6 |
| ✓ | ✗ | ✗ | 66.2 | 83.4 | 59.4 | 92.7 | 62.8 | 88.1 |
| ✗ | ✓ | ✗ | 68.1 | 77.9 | 61.5 | 92.1 | 64.8 | 85.0 |
| ✓ | ✗ | ✓ | 66.6 | 82.6 | 60.8 | 92.5 | 63.7 | 87.6 |
| ✗ | ✓ | ✓ | **72.2** | **73.6** | **62.6** | **92.1** | **67.4** | **82.9** |

**Table 6: The ablation study of MSC Loss and DME on Real to Real Benchmark.**

| ULIP Loss | MSC Loss | DME | SR3(easy) | | SR2(med) | | SR1(hard) | | Avg | |
|---|---|---|---|---|---|---|---|---|---|---|
| | | | AUROC↑ | FPR95↓ | AUROC↑ | FPR95↓ | AUROC↑ | FPR95↓ | AUROC↑ | FPR95↓ |
| ✗ | ✗ | ✗ | 56.4 | 88.8 | 55.8 | 90.6 | 52.3 | 99.1 | 54.8 | 92.8 |
| ✓ | ✗ | ✗ | 56.4 | 89.8 | 56.8 | 89.9 | 51.9 | 98.3 | 55.0 | 92.7 |
| ✗ | ✓ | ✗ | 58.1 | 88.7 | 58.2 | 89.9 | 55.9 | 99.8 | 57.4 | 92.8 |
| ✓ | ✗ | ✓ | 77.8 | 79.8 | 70.2 | 83.1 | 71.0 | 98.3 | 73.0 | 87.1 |
| ✗ | ✓ | ✓ | **90.1** | **63.8** | **72.2** | **78.1** | **79.5** | **97.4** | **80.6** | **79.8** |

sets: SN1, SN2, and SN3. The ten categories of ModelNet40 corresponding to ScanObjectNN are divided into SR1 and SR2, while the remaining five categories in ScanObjectNN form SR3.

*4.3.2* **Experiment Results**. When conducting novelty detection across the three benchmarks, we set the scoring coefficients w to 0.35, 0.3, and 0.15 respectively. Tab. 1 presents the results of the *Synthetic benchmark*, where our approach outperforms all other methods on SN1(hard), achieving top-1 performance. In detail, our approach demonstrates a 2.0% increase in AUROC and a reduction of 5.5% in FPR95 compared to DGCNN + cosine proto. In SN2(med) and SN3(easy), although not reaching the top-1, our method enables the only unsupervised method VAE to show competitive results. The results of our method on the *Synth to Real benchmark* are provided in Tab. 2. Due to the domain shift, performance degradation was observed in all methods. However, despite this challenge, we managed to further reduce FPR95 by 0.9% on SR1 (easy). In the *Real to Real benchmark*, the authors of 3DOS did not evaluate VAE. Hence, we supplemented this result and included our method in Tab. 3 for comprehensive comparison. Notably, our method achieves a 1.9% increase in AUROC on SR1 (hard) compared to the previous state-of-the-art method PointNet++ + cosine proto. These performance improvements are attributed to the rich information brought by pretraining as well as the capability of DME to uncover hidden novel patterns. Regarding other tasks, our method fell short of achieving the SOTA. This is because VAE[35] is the only method trained without the use of labels, unlike other methods evaluated in 3DOS[3]. Despite utilizing label information during pretraining, aligning multimodal features alone did not provide the 3D encoder with discriminative power comparable to methods trained through classification. Although our method did not achieve the state-of-the-art in every task, it significantly improved the performance of the original VAE across all tasks.

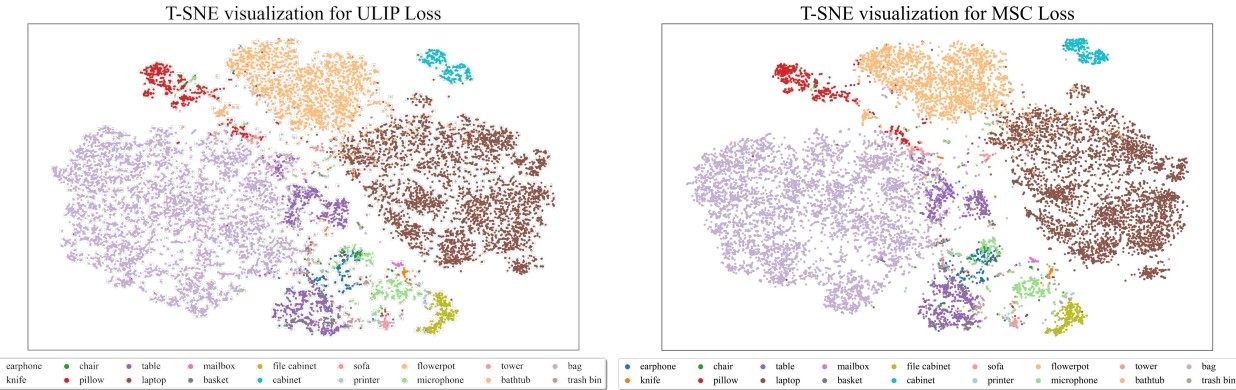

**Figure 3: The T-SNE visualization of ULIP Loss and MSC Loss. As illustrated, the MSC Loss enables samples from the same category to form denser clusters in the feature space.**

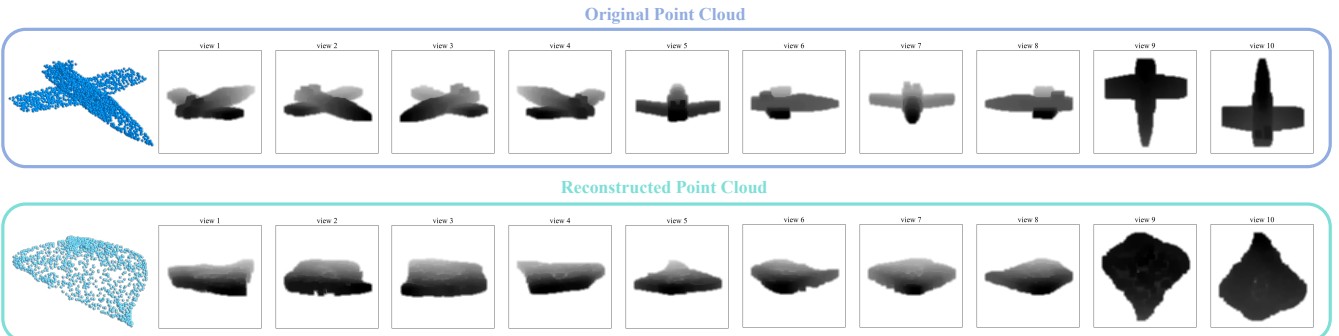

**Figure 4: The comparison of novel point cloud sample before and after reconstruction. Despite the chamfer distance indicating that it is a normal sample, the difference between its depth maps reveals its novel nature.**

## 4.4 Ablation Study

In order to investigate the exact contributions of MSC Loss on the pretraining phase and DME on novelty detection, independent ablation studies of the two modules are conducted in our approach. With the aim of evaluating model performance in novelty detection, AUROC and FPR95 metrics are utilized as quantitative evaluation criteria across three benchmarks. The comparison results across the three benchmarks are respectively illustrated in Tab. 4, Tab. 5, and Tab. 6.

*4.4.1 **ULIP Loss vs. MSC Loss**.* First, we conducted an ablation study to explore the impact of using different loss functions during pretraining. Our findings suggest that performing coarse-grained feature alignment during the pretraining stage can enhance the model's performance in subsequent novelty detection tasks. However, with the incorporation of MSC Loss, the model exhibited further improvements across different tasks on the three benchmarks, achieving average increases of 0.7%, 2.0%, and 2.4% in AUROC, respectively. This is attributed to MSC Loss leveraging label information to more effectively cluster data of the same category in the feature space, thereby enhancing the discriminative capability of the 3D encoder. As illustrated in Fig. 3, this approach enables data from the same category to acquire more similar features, thus facilitating the formation of denser clusters.

*4.4.2 **non-DME vs. DME**.* After demonstrating the effectiveness of MSC Loss, we show the role of DME. By applying DME in the novelty detection task, the performance across different tasks on the three benchmarks significantly improved based on models pretrained with MSC Loss. On average, the AUROC increased by 7.4%, 2.6%, and a remarkable 23.2%, respectively. DME utilizes the information provided by the depth maps of projected point clouds to effectively identify unseen samples with smaller reconstruction errors. As illustrated in Fig. 4, despite the reconstructed point clouds having smaller Chamfer Distance, their depth maps exhibit noticeable disparities. Through this simple yet effective metric, we can uncover latent novel samples and detect them, thereby significantly improving the performance of novelty detection.

## 5 Conclusion

In this paper, we introduce UniL, a multimodal pretraining framework tailored for 3D novelty detection. During pretraining, we propose the MSC Loss to better assist the 3D encoder in feature representation by leveraging label information. In the novelty detection phase, we utilize DME to measure the disparity between depth maps projected before and after reconstruction of point clouds, thus further enhancing the performance of unsupervised VAE. Through extensive experiments on the benchmark proposed in 3DOS[3], we validate the efficacy of our approach in 3D novelty detection tasks and provide valuable insights for future research.

## Acknowledgments

This work was supported by the National Natural Science Foundation of China (61906036) and the Big Data Computing Center of Southeast University.

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
