# OpenReview forum: "UniL: Point Cloud Novelty Detection through Multimodal Pre-training"
_acmmm.org/ACMMM/2024/Conference — MM2024 Poster_

### Official Review · Reviewer_zykk · 2024-05-24

**Rating:** 2
**Confidence:** 2

**Summary:**

This paper proposed a Unified Learning Framework for facilitating novelty detection to address the impact of 3D data scarcity on detection accuracy. In addition, authors introduced a new Multimodal Supervised Contrastive Loss (MSC Loss) and Depth Map Error(DME).

**Strengths:**

Being able to find key issues in 3D visual understanding and give solutions. The performance in the experimental data seems to be good.

**Limitations:**

1.The methodology of this paper is not innovative enough and the network is highly approximate to ULIP.
2.In pre-training, due to the limited amount of data, catastrophic forgetting can occur if the image and text encoders of CLIP are updated. When applying ULIP to downstream tasks, this leads to a significant performance degradation, and the authors' choice to freeze the weights of CLIP's visual and text encoders seems to solve the catastrophic forgetting. Please ask the authors what approach they utilize to address the problem of error accumulation in continuous learning.
3.The authors state that the MSC Loss enables samples from the same class to form denser clusters in the feature space, but the difference between the ULIP Loss and the MSC Loss is not that great from the visualization in Figure 3, which looks more like homologous scatter aggregation.

**Suitability:**

3

---

### Official Review · Reviewer_sGZN · 2024-05-27

**Rating:** 4
**Confidence:** 2

**Summary:**

This paper proposes a new Unified Learning Framework(UniL) for novelty detection. It argues that the scarcity of 3D data may lead to training bias and provides a more efficient 3D point cloud encoder trained by fusing 2D deepth images, texts and 3D point clouds. For novelty detection, built on unsupervised VAE, this paper designs a new MSC Loss to improve the compactness of clustering. A depth map error is additionally introduced to novelty score, reducing the error caused by potential distribution consistency. The effectiveness of the proposed method is verified by experiments, and the effectiveness of MSC Loss and DME is verified by ablation experiments.

**Strengths:**

- This article is very well written and clearly illustrated.
- For scarce point cloud data, this article proposes to use image and text modalities to enhance the encoding of point clouds.

**Limitations:**

- Although the multimodal encoder is able to encode 3D point clouds better, it would be useful to provide  the experimental validation of the comparison between proposed encoder and other encoders.
- The method proposed in this article can far outperform the original unsupervised VAE method, but still falls short of other supervised methods.

**Suitability:**

3

---

### Official Review · Reviewer_9S7K · 2024-06-02

**Rating:** 4
**Confidence:** 3

**Summary:**

This paper explores anomaly detection in 3D objects based on multimodal learning. This paper first propose the UniL framework to extract features from multi-view images, text descriptions, and 3D objects, and then use contrastive learning to align information between different modalities with the proposed Multimodal Supervised Contrastive Loss (MSC Loss) for pre-training a robust 3D visual encoder. Then it trains a Variational Auto-Encoder (VAE) based on the pre-trained 3D visual encoder to reconstruct the point cloud of the 3D object. Furthermore, this paper uses VAE and the proposed Depth Map Error (DME) for anomaly detection. Various experimental results basically prove the effectiveness of the proposed methods.

**Strengths:**

1. The writing and presentation of this paper is clear, the structure is reasonable, and the figures and tables are clearly presented.

2. The motivation of this paper is reasonable. Multimodal information, especially 3D information, needs to be learned and aligned.

3. The method proposed in this paper is novel and relatively complete, and seems to be able to represent 3D objects with depth information well.

4. The proposed method is evaluated on multiple benchmarks.

**Limitations:**

I still have some concerns about this paper:

1. For the proposed UniL, the paper does not conduct much exploration on the design of text input. Are these natural language-based inputs robust to the entire model? More ablation experiments are better.

2. For training VAE, I think it is very important to get good point cloud synthesis capabilities. For the output of the decoder, can their depth information be synthesised well? You should have some analysis here, which is very important for the proposed Depth Map Error (DME) in the next step.

3. For anomaly detection, we have more common metrics to represent anomaly scores, such as energy score, or knn distance. Will the proposed Depth Map Error (DME) be better than them?

Additionally, for the pre-trained multimodal feature space, although the experimental results are evaluated on multiple benchmarks, I am concerned that the alignment of the three modalities is difficult to generalize sufficiently, and I would like to get some more discussion here.

**Suitability:**

3

---

### Official Review · Reviewer_sQox · 2024-06-02

**Rating:** 3
**Confidence:** 2

**Summary:**

Compared to 2D data, the scarcity of available 3D data leads to the suboptimal representation of trained models. To address this challenge, authors propose a Unified Learning Framework (UniL) which aligns visual, textual, and 3D features within the same feature space for novelty detection.

**Strengths:**

The paper is well-organized and easy to follow.
The experiments are extensive.
The idea that proposes CLIP in novelty detection is interesting.

**Limitations:**

The motivation for detailed implementations, e.g., MSC Loss, and DME, is unclear.
The novelty is limited, as  Realistic Projection and contrastive loss are utilized in existing works. The author should highlight the novelty.
The performances do not reach SOTA, compared with PN2+Cosine Proto.

**Suitability:**

3

---

### Meta-Review · Area_Chair_rnPt · 2024-07-03

**Recommendation:** Accept (Poster)
**Confidence:** 4

**Metareview:**

The paper presents a well-organized and clearly written study on novelty detection using CLIP. The motivation is reasonable, focusing on learning and aligning multimodal information, particularly 3D information. The proposed method is novel, relatively complete, and appears effective in representing 3D objects with depth information. The method is evaluated on multiple benchmarks, with good performance in experimental data. The paper also proposes using image and text modalities to enhance the encoding of point clouds for scarce data, addressing key issues in 3D visual understanding.  All reviewers support accepting this paper, recognizing its novelty and clear writing. The authors address some concerns raised by reviewers during the rebuttal. The ACs agree with the reviewers and thus suggest accepting this paper.